# Carotenoids as Novel Therapeutic Molecules Against Neurodegenerative Disorders: Chemistry and Molecular Docking Analysis

**DOI:** 10.3390/ijms20225553

**Published:** 2019-11-07

**Authors:** Johant Lakey-Beitia, Jagadeesh Kumar D., Muralidhar L. Hegde, K.S. Rao

**Affiliations:** 1Center for Biodiversity and Drug Discovery, Instituto de Investigaciones Científicas y Servicios de Alta Tecnología (INDICASAT AIP), Clayton, City of Knowledge 0843-01103, Panama; jolivel.l@gmail.com; 2Department of Biotechnology, Sir M. Visvesvaraya Institute of Technology, Bangalore 562157, India; jk4research@gmail.com; 3Department of Radiation Oncology, Houston Methodist Research Institute, Houston, TX 77030, USA; mlhegde@houstonmethodist.org; 4Center for Neuroregeneration, Department of Neurosurgery, Houston Methodist, Houston, Texas 77030, USA; 5Weill Medical College of Cornell University, New York, NY 10065, USA; 6Center for Neuroscience, Instituto de Investigaciones Científicas y Servicios de Alta Tecnología (INDICASAT AIP), Clayton, City of Knowledge 0843-01103, Panama

**Keywords:** Alzheimer’s disease, Amyloid-β aggregation, carotenoid, apocarotenoid, biosynthesis, molecular docking analysis, structure-activity relationship

## Abstract

Alzheimer’s disease (AD) is the most devastating neurodegenerative disorder that affects the aging population worldwide. Endogenous and exogenous factors are involved in triggering this complex and multifactorial disease, whose hallmark is Amyloid-β (Aβ), formed by cleavage of amyloid precursor protein by β- and γ-secretase. While there is no definitive cure for AD to date, many neuroprotective natural products, such as polyphenol and carotenoid compounds, have shown promising preventive activity, as well as helping in slowing down disease progression. In this article, we focus on the chemistry as well as structure of carotenoid compounds and their neuroprotective activity against Aβ aggregation using molecular docking analysis. In addition to examining the most prevalent anti-amyloidogenic carotenoid lutein, we studied cryptocapsin, astaxanthin, fucoxanthin, and the apocarotenoid bixin. Our computational structure-based drug design analysis and molecular docking simulation revealed important interactions between carotenoids and Aβ via hydrogen bonding and van der Waals interactions, and shows that carotenoids are powerful anti-amyloidogenic molecules with a potential role in preventing AD, especially since most of them can cross the blood-brain barrier and are considered nutraceutical compounds. Our studies thus illuminate mechanistic insights on how carotenoids inhibit Aβ aggregation. The potential role of carotenoids as novel therapeutic molecules in treating AD and other neurodegenerative disorders are discussed.

## 1. Introduction

The onset of neurodegenerative disease is a consequence of abnormal conformational/structural changes in proteins [1]. Alterations in the biochemical homeostasis of the nervous system due to metal deposition, oxidative stress, disturbances in cholinesterase, and amyloid and tau plaque formation are highly associated with the initiation and progress of neurodegenerative disease [2,3,4,5,6,7,8,9,10,11]. In addition, the accumulation of misfolded amyloidogenic proteins in the central nervous system (CNS) has been linked to progressive neurological impairment [5]. AD, Parkinson’s disease (PD), Huntington’s disease (HD), and amyotrophic lateral sclerosis (ALS) are the most devastating neurodegenerative diseases, and all of these conditions affect normal behavior [7,8,12]. AD is the most prevalent worldwide with more than 45 million aging people suffering from this condition today and the number of cases expected to triple by 2050 [2]. According to the Aβ hypothesis, Aβ peptide is considered the main trigger of AD pathogenesis due to reduced Aβ clearance [13,14]. In the human brain, Aβ fibrils are considered the hallmark of AD and are found in the hippocampus and neocortical regions [13]. Drug discovery for AD faces two major challenges, namely the lack of an animal model which represents all events that occur in a patient with AD and the absence of a reliable biomarker that enables monitoring of disease progression [15,16,17,18]. 

Natural products, such as polyphenols, carotenoids, and other compounds, play an important role in the prevention of neurodegenerative diseases [19,20,21,22,23]. Polyphenols, such as resveratrol, epigallocatechin gallate (EGCG), and curcumin, constitute a large family of compounds that have demonstrated neuroprotective activity [6]. The biggest challenges with these compounds are their stability and bioavailability. However, structural modification of polyphenols has been demonstrated to improve their biological activity [24]. 

Carotenoids are another type of natural product reported to play a role in the prevention of brain disorders [25,26]. These molecules are natural products distributed widely in plants, animals, and microorganisms [27], and more than 750 carotenoids exist in nature. These compounds are responsible for the yellow, orange, and red color in various plants and animals (birds, reptiles, fish), and the color is indicative of the type of carotenoid [28,29,30,31,32,33]. Carotenoids are secondary metabolites produced by enzymatic reactions, and these reactions generate different carotenoids with biological importance [34,35]. Carotenoids play a variety of important functions in plants, animals, and in the human brain. In plants, carotenoids are synthetized de novo where most are enriched with a small quantity of its biosynthetic precursor [36]. The biosynthesis of carotenoids is mediated by different enzymes and occurs in the plastids. They have functions such as light harvesting in the photosynthetic membranes and protection of the photosynthetic apparatus against photo-oxidation [34,37]. Carotenoids also act as precursors of the phytohormone abscisic acid (ABA) [38] and are involved in the process of pollination as attractors of pollinators and seed dispersal agents [34]. Some animals can synthetize carotenoids, but most depend on dietary intake. In humans, these compounds have several therapeutic properties, including antioxidation, prevention of neuro-inflammation, anti-cancer, anti-diabetes, prevention of cardiovascular disease, and potentially the prevention and treatment of neurodegenerative disorders [39,40,41,42,43]. The antioxidant property of carotenoids is the main therapeutic characteristic of this compound where the conjugate polyene chain plays a crucial role in the reduction of free radicals [43,44,45,46]. This paper is focused on the chemistry of carotenoids and, through molecular docking analysis, proposes plausible mechanisms by which they may inhibit Aβ aggregation. 

## 2. Chemistry of Carotenoid Compounds 

Carotenoids belong to a class of terpenoid compounds with a characteristic conjugated polyene chain composed of eight C_5_ isoprenoid units (Figure 1). The trans-carotenoid isomer is most predominant in nature [47,48,49,50,51]. The large polyene chain enables light absorption in the range of 450–550 nm of the visible spectrum [52]. The *cis*- and trans-carotenoid isomers have varying melting points, with *cis*-isomers having a lower value than trans-carotenoids due to the decreased tendency of *cis*-isomers to crystallize [28,53]. Meanwhile, *cis*-carotenoids can absorb light at a shorter wavelength, with a lower absorption coefficient [54]. The soluble property of carotenoids is based on their hydrophobic chemical structure, which makes them insoluble in water and dimethylsulfoxide, but favors their solubility in organic solvents such as ethyl ether, chloroform, acetone, ethyl acetate, and ethanol [55]. Most carotenoids are lipophilic with the ability to cross the blood-brain barrier [56]. Carotenoids are considered “molecular rivets” since the width of this type of compound is the same as that of the phospholipid bilayer which confers stability and rigidity to the phospholipid membrane [57,58]. 

Carotenoids are classified into two groups, namely carotenes and xanthophylls, both of which can be acyclic or cyclic compounds [59,60,61]. Carotenes have carbon and hydrogen atoms. The most common acyclic carotene is lycopene, which is found in tomato, red-fleshed papaya, and guava [55]. The most common cyclic carotene is β-carotene, which is found mainly in carrot, apricot, and mango [55]. Other cyclic carotenes are β-zeacarotene, α-zeacarotene, γ-carotene, δ-carotene, and α-carotene. Xanthophylls contain carbon, hydrogen, and oxygen atoms in their structure forming hydroxyl, epoxide, and keto groups [59,60]. Xanthophylls are formed from enzymatic reactions involving hydroxylases, epoxidases, or synthases. Carotenols are xanthophylls that possess a hydroxyl group (e.g., β-cryptoxanthin, zeaxanthin, and lutein), while epoxycarotenoids are xanthophylls with an epoxide group (e.g., antheraxanthin, violaxanthin, neoxanthin, β-carotene-5, 6-epoxide, β-carotene-5, 6, 5’,6’-diepoxide, and β-Cryptoxanthin-5,6-epoxide) (Figure 2). Ketocarotenoids, such as astaxanthin, fucoxanthin, capsanthin, and capsorubin, are xanthophylls with a keto group [62]. 

## 3. Biosynthesis of Carotenoids

Carotenoid synthesis can begin with the mechanism (1–4’) head-to-tail condensation of three isopentenyl diphosphates (IPP) and one dimethylallyl diphosphate (DMAPP) to produce the isoprenoid compound known as geranylgeranyl diphosphate (GGPP) [63]. The olefinic double bond of IPP nucleophilically attacks the carbocation produced through release of the allylic diphosphate group [63]. This reaction is catalyzed by GGPP synthase (GGPPS), and this enzyme has a mechanism of cyclic condensation: ionization-condensation-elimination [48,63]. GGPP is the precursor of different types of compounds such as carotenoids, plastoquinones, tocopherols, chlorophyll side chains, and phytohormones [63]. The head-to-head coupling of two GGPP catalyzed by phytoene synthase (PSY) produces a dimerized compound known as phytoene [34,64]. A phytoene compound is characterized by the presence of three conjugated double bonds in a chain. 

The production of carotenoids starts with phytoene synthesis, and this process is considered the rate-controlling reaction in the pathway [63]. The biosynthesis of phytoene is the first committed step in the desaturation process which leads to the formation of the conjugate double bond that is characteristic of carotenoids (Figure 2). The enzyme phytoene desaturase (PDS) introduces two conjugated double bonds into the long chain, forming a phytofluene compound. Then, the desaturation reaction gives rise to another two conjugated double bonds, producing ζ-carotene. The poly-cis-carotenoid compound generated in this biosynthetic process is converted into trans-carotenoid by carotene isomerase (CRTISO) or ζ-carotene isomerase (ZISO). The neurosporene compound is biosynthesized by ζ-carotene desaturase (ZDS), which introduces two conjugate double bonds into the chain [64]. This desaturase enzyme continues the enzymatic process to form the lycopene compound by introducing two conjugate double bonds. The cyclization of lycopene was proposed by Porter and Lincoln, who hypothesized that cyclization can occur on one or both sides of the carotenoid to generate different type of compounds [63]. Two different rings, such as β- or ε-ionone rings, are generated from lycopene cyclization. Usually, the main modification of the carotenoid occurs on the end, and only a few modifications occur in the long chain of the carotenoid. The enzyme responsible for lycopene cyclization is lycopene cyclase (LCY), which is found in plants and cyanobacteria. The type of carotenoid is determined by the LCY subtype, namely β–ionone (LCYB) rings, ε-ionone (LCYE) rings, or bifunctional β/ε-ionone (LCYB/E) rings. An unsubstituted β-ionone ring is considered a source of vitamin A, a molecule important for human health. In the presence of LCYB, lycopene is transformed into γ-carotene. Meanwhile, lycopene in the presence of LCYE is transformed into δ-carotene. Both compounds contain a six-member-ring at the end of the chain. LCYB can transform γ-carotene into β-carotene, a compound with two six-member rings at the end of the chain. A similar transformation occurs with δ-carotene, which is transformed into α-carotene. The addition of hydroxyl and keto groups to carotenoids can be achieved through two different classes of enzymes, ferredoxin-dependent nonheme diiron enzymes (HYD) and enzymes to belonging to the cytochrome P450 family [63]. The first is a type of β-hydroxylase that is responsible for β-cryptoxanthin and zeaxanthin formation. Cytochrome P450 has a novel ε-hydroxylase that is responsible for hydroxylation of the ε-ionone ring to form zeinoxanthin. The enzyme capsanthin-capsorubin synthase converts epoxycarotenoids into κ-cyclic carotenoids such as capsanthin and capsorubin [65]. Epoxidation reactions also occur in carotenoids through epoxidases such as zeaxanthin epoxidase (ZEP), which is responsible for antheraxanthin and violaxanthin formation. 

Apocarotenoids have a structure similar to carotenoids but without C_40_ atoms (Figure 3). An apocarotenoid is considered a product of oxidative cleavage of carotenoids, and the most well-known apocarotenoids are Apo-12’-capsorubinal and bixin [66,67]. 

The five important reactions involved in carotenoid biosynthesis are desaturation (1), cyclization (2), hydroxylation (3), epoxidation (4), and epoxide-furanoxide rearrangement (5) (Figure 2) [55]. The desaturation reaction (Figure 4) is essential for the formation of conjugate double bonds in the chain of the carotenoid. Formation of the lycopene compound starts with desaturation of phytoene, which contains three conjugated double bonds, and then proceeds with the formation of phytofluene, which contains five conjugated double bonds, ζ-carotene, which possesses seven conjugated double bonds, neurosporene with nine conjugated double bonds, and finally lycopene, which contains 11 conjugated double bonds [55,68]. Because the flavin cofactor can accept one or two electrons, four mechanisms have been reported to demonstrate how phytoene desaturation proceeds, namely through single electron transfer, the formation of hydride from substrate, transfer to flavin adenine dinucleotide (FAD), hydrogen atom transfer, and nucleophilic attack [68]. The cyclization reaction (Figure 4) is important in ring formation of the carotenoid. Lycopene is the important carotenoid which starts the cyclization process [69]. The reaction mechanism consists of attacking the alkene in the C_2_ position of the lycopene to generate an acid-producing carbocation in the lycopene, which is stabilized by deprotonation of hydrogen in C_4_, C_18_, or C_6_ to yield a β-end-group, γ-end-group, or ε- end-group (Figure 3). Two types of rings can be formed, such as the β-ionone ring, at the end-group of both sides (e.g., β-carotene) [69]. The ε-ring can be formed at one side of the compound such as in α-carotene [69]. 

The hydroxylation reaction (Figure 5) is quite common in β- or ε-ionone rings in carotenoids. The addition of this group to the ring causes a loss in vitamin A properties [63]. Xanthophyll functions in preventing damage caused by photo-oxidation. The hydroxylation reaction is carried out by two classes of enzymes, namely ferredoxin-dependent nonheme diiron enzymes (HYD) and enzymes of the cytochrome P450 family [70,71]. A plausible mechanism of carotenoid hydroxylation by HYD has been proposed based on studies of alkane mono-oxygenase and ʟ-*p*-amino phenylalanine hydroxylase, where the first step consists of binding of one oxygen molecule to an iron cluster to form the peroxodiferric intermediate. This step is followed by hydrogen abstraction of the carotenoid by a potent oxidant (Q), resulting in the formation of a carotenoid radical that interacts with the iron complex to hydroxylate the carotenoid [70,71,72]. The epoxidation reaction (Figure 5) of carotenoids can lead to structural modification of the β-ionone ring at the 5,6-position (e.g., antheraxanthin is epoxidized at one site by zeaxanthin epoxidase (ZEP) [38,69]. However, the epoxidation reaction is a reversible reaction in the presence of violaxanthin de-epoxidase (VDE), forming the xanthophyll cycle which plays an important role in the regulation of photosynthetic energy conversion [73]. In the epoxide-furanoide rearrangement reaction (Figure 5), the epoxide carotenoid can undergo epoxide-furanoxide rearrangement in the presence of acid to produce a different type of end-group. The type of end-group is determined by the carbocation position and hydrogen abstraction by a base (B^-^), such as the *k*-end-group, γ-end-group, or ε-end-group, which is produced when the carbocation is in the C_5_ position. Meanwhile, the epoxide-furanoxy rearrangement known as 5,8-epoxide carotenoid occurs when the carbocation is in the C_6_ position [69]. Capsanthin-capsorubin synthase (CCS) is the enzyme responsible for *k*-end-group formation, where pinacol rearrangement occurs to form a new family of carotenoids [67]. 

## 4. Types of Carotenoid Derivatives 

The synthesis of carotenoid derivatives with hydrophilic properties are a good alternative for incorporating this type of molecule with multiple therapeutic properties in aqueous medium [74,75]. Carotenoids have an important nutritional value since most of them are a source of vitamin A, antioxidant molecules with anti-carcinogenic properties, and the ability to prevent cardiovascular disease [41,45]. Synthesizing hydrophilic carotenoid derivatives is important due to their use as therapeutic molecules that improve the biological activity of carotenoids. Another reason to produce synthetic carotenoid derivatives is their increased water solubility. This property allows carotenoids to be widely used in the food industry as natural dyes. Several types of carotenoid derivatives have been synthetized, including reduced carotenoids, carotenoid epoxides, carotenoid esters, carotenoid glycosides, and PEGylated carotenoids (Figure 6) [75]. Below, we briefly describe the types of carotenoid derivatives synthesized with hydrophilic properties.

*Reduced carotenoids:* Similar to cryptocapsin and capsorubin, carotenoids are difficult to isolate or detect. However, the increased polarity of their reduced form can be utilized in their detection [76]. The reagent NaBH_4_ in EtOH or EtOH/benzene is used to reduce the keto group of the carotenoid to generate two stereoisomeric alcohols (Figure 6) [76]. *Carotenoid epoxides:* Carotenoids are a source of vitamin A; however, when the β-ionone ring is epoxidized, the pro-vitamin A activity is lost [65,77]. This epoxidation is prepared using monoperoxyphthalic acid to produce two types of epoxides that can be separated through a chiral column (Figure 6) [77,78]. *Carotenoid esters*: Carotenoids, with the exception of bixin and crocin, are poor water-soluble molecules. The synthesis of carotenoid derivatives with water-soluble characteristics enables the therapeutic properties of this type of compound in aqueous solutions (e.g., Cardax) [75,79]. Esterification of carotenoids is most frequently performed using succinate, phosphates, and lysinates [74,75,80]. Succinate is prepared using succinic anhydride with 4-dimethylaminopyridine (DMAP) in dichloromethane (CH_2_Cl_2_) [39,80,81]. Meanwhile, phosphate is produced using lutein with dibenzyl phosphoroiodidate in CH_2_Cl_2_ [82]. Finally, preparation of the astaxanthin-amino acid conjugate can be achieved using *N*^αε^-bis-(tert-butoxycarbonyl)-L-Lysine (BocLys(Boc)OH) with DMAP in CH_2_Cl_2_ and 1,3-diisopropylcarbodiimide (DIC), [83]. These types of carotenoid derivatives have been demonstrated to exhibit increased solubility in polar solvents when they are converted to sodium salt [82]. 

*Carotenoid glycosides*: This type of carotenoid derivative is found in nature in some microorganisms that are characterized by their resistance to heat, specifically the *Thermus* species. Thus, this type of compound is known as a thermoxanthin [75]. The presence of glycoside makes it more water soluble and amphiphilic. Yamano, et al. achieved synthesis of thermocryptoxanthin and thermozeaxanthin [84]. Glucosidation can be performed by combining glucosyl bromide with silver triflate (AgOTf) in some cases in the presence of N,N,N’,N’-tetramethylurea as a proton acceptor (Figure 6) [84]. *PEGylated carotenoids*: A polyethyleneglycol (PEG) conjugate is widely used for its pharmacokinetic properties and solubility in aqueous solution [39,74]. This type of carotenoid derivative is synthetized by coupling succinate with polyethylene glycols using dicyclohexylcarbodiimide (DCC) with DMAP in CH_2_Cl_2_ [81,85]. Another type of carotenoid derivative is produced by the azide-alkyne click reaction, which is well known for increasing water solubility (Figure 6) [75,86].

## 5. Neuroprotection by Carotenoids through Modulation of the App Pathway

Aβ peptide is formed by cleavage of amyloid precursor protein (APP), which is catabolized by secretases (α, β, and γ secretase), to form non-amyloidal and/or amyloidal products [87,88,89]. APP is a multidomain glycoprotein with signal peptide regions located at residues 1–17, two copper-binding sites at amino acids (aa) 124–189, and an E2 domain 376 where chromosome 21 produces eight isoforms of APP (695–770). The major isoform is APP 695. When the N-terminal region of APP is cleaved by β-secretase, secreted APP fragment beta (sAPPβ) is formed and γ-secretase transforms C-terminal fragment beta (CTFβ) in Aβ at the copper-binding site [90]. The non-amyloidogenic α-secretase belongs to the family of zinc metalloproteinases that cleaves APP at Lys687. Amyloidogenic aspartyl protease β-secretase cleaves at Met671, while γ-secretase cleaves at Val711 and Ala713 to form Aβ_40_ and Aβ_42_, respectively [90,91,92,93].

Several studies have demonstrated that carotenoids can inhibit Aβ aggregation [45,56,94,95,96]. Carotenoids such as lutein, lycopene, astaxanthin, fucoxanthin, cryptocapsin, and the apocarotenoid crocin have been demonstrated to inhibit Aβ_42_ aggregation (Figure 7). Lutein, a hydroxylated carotenoid present in apricot extract, exhibits a strong inhibition of Aβ_42_ fibril formation [56]. Katayama, et al. compared lutein with zeaxanthin, β-cryptoxanthin, β-carotene, and α-carotene, and found that lutein has the highest activity. The authors hypothesized that the number of hydroxyl groups plays an important role in the inhibition of Aβ aggregation [56]. Lycopene, an acyclic carotene found in tomatoes with a highly conjugated structure, has been shown to exhibit multiple benefits including inhibition of Aβ_42_ formation [94,95,96]. Wang, et al. found that lycopene can reduce Aβ aggregation and inhibit the inflammatory response by suppressing microglia activation and increasing antioxidant enzyme activities [94]. Crocin, an unusual polar apocarotenoid found in *Crocus sativa* stigma, possesses an inhibitory activity similar to Aβ [43,97]. Papandreou, et al. found that trans-crocin-4 is the main apocarotenoid present in the perennial herb of the Iridaceous family [43]. Astaxanthin, a ketocarotenoid found in crustacean shells, is also considered to be a highly antioxidant molecule [62,98]. Lee, et al. found that astaxanthin can inhibit nitric oxide production in vitro and in vivo [98]. Fucoxanthin, an allene carotenoid found in marine microorganisms, have multiple therapeutic properties such as reduction of oxidative stress, decreased inflammation, decreased obesity effect, and inhibition of Aβ assembly [99,100]. Xiang, et al. demonstrated that fucoxanthin could inhibit Aβ assembly and that this inhibition is reinforced by hydrophobic interaction between this carotenoid and the Aβ peptide [100]. A recent study showed that fucoxanthin could inhibit β-secretase by hydrogen bonding with this enzyme [101]. Inhibition of this enzyme is another neuroprotective activity of carotenoids, and occurs by blocking the amyloidogenic pathway via inhibition of the initiation of APP cleavage to produce Aβ peptide [101]. Cryptocapsin is a ketocarotenoid with a non-conventional keto-kappa group found in *Pouteria sapota* [102,103]. We found that this carotenoid could potently inhibit Aβ_42_ aggregation whereas cryptocapsin-5,6-epoxide, which is found in the same fruit, exhibited only a minor activity. Both carotenoids were compared with zeaxanthin, and the data demonstrated that cryptocapsin and zeaxanthin showed potent inhibition of Aβ_42_ aggregation [25]. These results led us to conclude that hydroxyl and keto groups increase their activity while an epoxide group reduces it [25]. 

Carotenoids are fascinating molecules that exhibit high antioxidant potential [104,105,106]. Lycopene, lutein, fucoxanthin, astaxanthin, cryptocapsin, and crocin are the most effective carotenoids in inhibiting Aβ aggregation. They have different structural characteristics, such as cyclic and acyclic structures, where the presence or absence of hydroxyl, epoxide, allenic, and keto groups is observed. 

Our question is, when the process of aggregation begins, how does the carotenoid inhibit Aβ aggregation? We observed that a long chain of conjugated double bonds is present in these carotenoids (Figure 7), and that this polyene chain is responsible for their antioxidant activity by stabilization of radical species [52,107]. Nevertheless, how does the carotenoid inhibit Aβ aggregation? We observed that these carotenoids are oriented perpendicular to the Aβ peptide. This leads us to infer that the long polyene chain may block stacking of the peptide. Most likely the carotenoid conformation can be reinforced by the hydrogen bonding interactions or π-π stacking interactions that occur between the polar residues of the Aβ peptide, such as Glu22, Asp23, and Gly25, and the functional groups of the carotenoid (e.g., mainly the hydroxyl groups) [25]. Based on this observation, we conducted molecular docking analysis using five carotenoids with different structures and evaluated them against Aβ peptide aggregation.

## 6. Molecular Docking Analysis of Carotenoid and Apocarotenoid against Aβ Aggregation

Cryptocapsin, lutein, fucoxanthin, astaxanthin, and bixin were the carotenoids selected for molecular docking analysis. Because cryptocapsin was the most active in our experimental results, this compound was selected as our standard. Our objective was to perform structural analysis of carotenoids to evaluate how they prevent Aβ aggregation. Despite the fact that a vast number of carotenoids exist in nature, these five were selected based on their structural differences to examine whether these variations play a crucial role in Aβ aggregation.

We used a computational structure-based drug design and evaluated the possible interaction modes between different forms of Aβ fibrillar species, including Aβ_42_ and Aβ_40_, with carotenoid ligands like astaxanthin, bixin, cryptocapsin, fucoxanthin, and lutein retrieved from the PubChem database [108]. The molecular docking experiments were performed using CDOCKER, a docking protocol developed by Discovery Studio 3.5 (Accelrys, San Diego, CA, USA). The three-dimensional (3D) solution and solid state NMR structures of U-shaped pentamer Aβ_17–42_ Protein Data Bank (PDB) identification (ID) were: 2BEG S-shaped Model Aβ_11–42_; PDB ID: 2MXU Disease-Relevant structure of Aβ_42_; PDB ID: 2NAO and Aβ_40_ derived from the brain of a patient suffering from AD; and PDB ID: 2M4J obtained from the RCSB PDB [109] (Figure 7).

The ligands and retrieved target PDB Aβ fibril NMR structures were minimized by using the conjugate gradient protocol and employing the CHARMm force field with Max Steps-2000 implemented in Discovery Studio 3.5 software (Accelrys, San Diego, USA).

The initial step was carried out by preparing different forms of Aβ fibrils and identifying the binding sites by using the Define and Edit binding site tool. The binding sites found are shown in Figure 8 as a set of points. Docking studies were carried out using the CDOCKER protocol under the Receptor-Ligand Interaction tools in Discovery Studio 3.5. CDOCKER is a powerful CHARMm-based docking method that has been shown to generate highly accurate docked poses [110]. 

The carotenoid ligands astaxanthin, bixin, cryptocapsin, fucoxanthin, and lutein (Figure 9) were docked onto the different forms of Aβ_42_ and Aβ_40_ fibrils. First, the Aβ fibril was held in a rigid conformation and the carotenoid ligands were allowed to flex during the refinement. Docking runs were adjusted to the 10–100 best outputs for evaluation of binding pose accuracy. Consequently, diverse poses are generated which adopted the random rigid body rotation and simulated annealing. Ligands displayed efficient docking with the generation of 10 conformers each. For each carotenoid ligand, the top 10 ligand binding poses were ranked according to their CDOCKER energies, which were calculated based upon the internal ligand strain energy and receptor-ligand interaction energy. The CDOCKER interaction signifies the energy of the non-bonded interaction that exists between the protein and ligand. It must be noted that, in both cases, greater CDOCKER energy and CDOCKER interaction energy values imply more favorable binding between the Aβ fibril and the ligand and therefore the best shape complementarity. The simulated docked complexes were studied based on the lowest binding affinity values (kcal/mol) and bonding interaction patterns (hydrogen, hydrophobic, and electrostatic). All ligand-protein interactions that exhibited stronger binding affinity and energies relative to individual targets are computed. 

It is interesting to understand that different types of structural polymorphism in the Aβ fibrils occur from the same polypeptide sequence in AD. These differences lead to variations in protofilament number, arrangement, and polypeptide conformations to numerous dissimilar patterns of inter- or intra-residue interactions. However, the most prevalent forms of Aβ peptides are Aβ_40_ and Aβ_42_ aa residues, with Aβ_42_ being the more toxic form over Aβ_40_. The higher toxicity of Aβ_42_ chains may be explained by their ability to adopt three-stranded S-shaped antiparallel β-strands as a cylindrical barrel/pore-like structure when assembling as fibrils. In contrast, U-shaped configurations can only be observed for Aβ_40_ peptides. Consistent with this, the patient-derived fibril structure of Aβ_40_ chains possesses a three-fold symmetry which cannot take an S-shaped form but rather a U-shaped configuration that differs from other fibril polymorphs, which are more significant than other polymorphs for the pathology of AD. Hence, the present computational study is also intended to understand and appreciate the binding modes of carotenoids and their effect on fiber stability and fiber toxicity with four different forms of Aβ. The initial outcomes of the binding site analysis revealed that all four Aβ models exhibited interesting known binding sites and different small and large binding regions with uninformed size and depth across the Aβ fibril surface, where small molecule/large ligand binding interactions can occur. Docking studies of astaxanthin, bixin, cryptocapsin, fucoxanthin, and lutein binding with the selected model showed promising interactions of the ligands with each docked fibril. The ligand-protein complex was evaluated for hydrogen bond interactions and CDOCKER interaction (kcal/mol−1) energies. 

Evaluating the interactions revealed that all the ligands exhibited strong interactions with residues such as Glu22, Val24, Gly25, Asp23, Ser26, Lys28, and Leu34, which participate in hydrogen bond formation (Table 1). Moreover, Ala21, Ile32, Gly33, Gly38, and Val39 participated through van der Waals interactions. Lutein showed the lowest CDOCKER energy (97.6659 kcal·mol−1), signifying greater interaction across 2BEG Aβ_42_ fibrils. Irrespective of the binding energies projected for the ligands, all exhibited a similar binding mode pattern. Based on the data presented in Table 1, carotenoid-mediated inhibition of Aβ aggregation can be ranked as lutein > bixin > cryptocapsin > astaxanthin > fucoxanthin.

Similarly, this analysis revealed important interactions between carotenoids and 2MXU Aβ_11–42_ (Table 2). All of these residues have been shown to interact with ligands via hydrogen bond formation while the remaining amino acids in the pocket display van der Waals interactions. Lutein (−60.6453 kcal/mol) exhibited the lowest CDOCKER energy. Regardless of their binding energies, all of the ligands exhibited a similar binding mode pattern with shared interacting residues such as Val12, His14, Ile32, Gly33, and Leu34. Based on the data in Table 2, carotenoid-mediated inhibition of Aβ aggregation can be ranked as lutein > cryptocapsin > fucoxanthin > astaxanthin > bixin.

Furthermore, the docking outcomes of the 2NAO-Disease-Relevant structure of Aβ_42_ with carotenoids are shown in Table 3. These data reveal that all of the ligands form stable hydrogen bonds and van der Waals interactions. The other amino acid residues across the ligands participated via van der Waals interaction. However, cryptocapsin (−84.343 kcal/mol) and lutein (−88.1515 kcal/mol) exhibited the lowest CDOCKER energy. While all ligands share common interacting residues like Ser8, Glu11, Lys16 across the binding site of 2NAO. Based on the data in Table 3, carotenoid-mediated inhibition of Aβ aggregation can be ranked as lutein > cryptocapsin > astaxanthin > fucoxanthin > bixin.

Carotenoids also interact with 2M4J-Aβ_40_ derived from the brain of a patient suffering from AD. As shown in Table 4, all of the ligands exhibited strong interactions with residues such as Glu28 and Gly37 through hydrogen bonding. The other amino acids participated by van der Waals interactions. Cryptocapsin (−101.782 kcal/mol) and lutein (−110.897 kcal/mol) exhibited the lowest CDOCKER energy. Based on the data in Table 4, carotenoid-mediated inhibition of Aβ aggregation can be ranked lutein > cryptocapsin > fucoxanthin > astaxanthin > bixin.

A comparison of the different highest CDOCKER energy values allowed us to propose a hypothetical percentage of inhibition of Aβ aggregation by these five carotenoids (Figure 9). We observed that lutein was the most powerful carotenoid against Aβ aggregation followed by cryptocapsin. Astaxanthin and fucoxanthin showed moderate inhibition of Aβ aggregation. However, bixin, an apocarotenoid, displayed the lowest inhibition of Aβ aggregation. Therefore, we propose that carotenoids have greater potential to inhibit Aβ aggregation than apocarotenoids.

According to published literature, ligands which bind to the Aβ_16–20_ region of Aβ_42_ are capable of disrupting Aβ fibril formation. Likewise, studies have revealed that amino acid residues 16 to 20 in Aβ_16–20_ are required for the self-association of Aβ peptides into Aβ fibrils. Aβ_16–20_ forms an antiparallel β-sheet structure through binding to the homologous regions of amino acids 17–21 or 18–22 [111,112]. Similarly, it has been shown that Aβ_16–20_ deletion, as the most important nucleation site, inhibits formation of the Aβ fibril [113]. Therefore, our docking analysis clearly indicates that ligands strongly interact in close contact with the 20–28 regions and hydrogen bonding at the Asp23-Lys28 salt bridge, resulting in disruption of the stability of the salt bridge and the oligomer structure as well as sheet breaking activity. Moreover, C-terminal residues such as Gly33, Met35, Val36, and Gly37 indicate that these ligands may disaggregate Aβ fibrils through their binding [111,113]. The data in Table 5 suggest that carotenoids and apocarotenoids can disrupt Aβ aggregation and the disaggregation pathway. Lutein can potentially inhibit oligomer formation due to interactions with residues 20–28 of the Aβ peptide, which are essential for oligomerization. Cryptocapsin has the potential to inhibit Aβ fibril formation due to interaction with residues 16–20. This carotenoid can also disaggregate Aβ fibrils through interaction with residues 33–37. Fucoxanthin is different from these carotenoids due to its potential to inhibit Aβ fibril formation, Aβ disaggregation, and oligomerization through interaction with residues 16–20, 24–28, and 33–37. Astaxanthin interacts with the Aβ peptide at amino acids 8–14, which are crucial for aggregation through interactions with metals such as Zn (II) and Cu (II) or Fe (II) and Al (III). This is a crucial metal-binding region [114]. However, we can hypothesize that its inhibition is based on disruption of hydrogen bond formation with residues 8–14 where this carotenoid has strong interactions with the polar residues of the Aβ peptide. Bixin, an apocarotenoid, has less contact with the Aβ peptide than carotenoids because it consists of half the number of carbons and possesses a cis conformation that reduces contact with the Aβ peptide. However, this apocarotenoid can interact with polar amino acids such as Glu11 and Glu22, Lys28, Gly33, and Leu34, indicating a potential for disrupting hydrogen bond formation with the Aβ peptide.

These results have allowed us to evaluate the inhibitory potential that carotenoids and apocarotenoids may possess. These data provide hints of their aggregation and disaggregation potential against Aβ formation. Some carotenoids can inhibit Aβ aggregation by interacting with residues in the regions of 8–14 and 16–20, which are essential for this procedure. On the other hand, there are carotenoids that can disaggregate Aβ fibrils by interacting with the 33–37 regions that are required for the fibrillation process. Nevertheless, there are carotenoids that can inhibit amino acids 23–28, which are essential for oligomerization. We also observed that there are carotenoids that may inhibit Aβ fibril aggregation and disaggregation.

Thus, we propose a plausible mechanism by which carotenoids inhibit Aβ aggregation and disaggregation. Our mechanism starts a proposal that the long chain of conjugated double bonds, characteristic of carotenoids, has a dual function. First, this polyene chain confers an antioxidant property to carotenoids by stabilizing radical species. Second, based on the planar structure of the polyene chain due to trans-configuration that is longer than the Aβ peptide, a perpendicular orientation may block self-aggregation of the peptide. The conformation of the carotenoid is reinforced by chemical interactions between the functional group of the carotenoid with polar residues of the Aβ peptide. For instance, all four carotenoids examined in this study share a similar polyene chain structure with trans-configuration. However, the apocarotenoid (bixin) has a short polyene chain with a *cis*-configuration. The carotenoid with a trans-configuration may have more potential to inhibit Aβ aggregation than a molecule with a *cis*-configuration. Another important difference that was observed is related to the quantity and positioning of hydroxyl groups within the molecule. For example, lutein, the most powerful carotenoid against Aβ aggregation, has two hydroxyl groups in the two β–ionone ring, specifically in the C_3_ position. Cryptocapsin possesses one hydroxyl group in the 6-oxo-kappa end-group, specifically in the C_3_ position. Astaxanthin has two hydroxyl groups in the C_3_ position; however, this carotenoid is predicted to have lower potential for inhibiting Aβ aggregation than lutein. Therefore, it is possible that the keto-kappa groups and configuration of hydroxyl groups play a role in the inhibition of Aβ activity. Fucoxanthin has two hydroxyl groups but in different positions, as well as an epoxide group and an allene group that could influence the activity. The apocarotenoid bixin demonstrated the importance of the double-bond chain length in Aβ inhibition. These preliminary results allow us to propose a structure-activity relationship (SAR) where the double-bond chain is crucial for contact with the Aβ peptide. However, the hydroxyl group in the hydrogen bond interaction, along with their position and configuration in the molecule, is also important. Finally, the presence of epoxide, allene, and keto groups in the molecule may also affect Aβ inhibition. 

In the published literature, little data exists showing the neuroprotective activity of carotenoids. While studies of these compounds have demonstrated their role in inhibiting the aggregation or disaggregation of Aβ fibrils, the underlying mechanism remains unknown. Thus, our proposed mechanism is based on two requirements: 1) blocking self-aggregation of the Aβ peptide by the length and configuration of the compound avoids self-assembly; and 2) the quantity and position of the hydroxyl groups are essential for interacting with the Aβ peptide/fibril through hydrogen bonding. Figure 10 is a schematic detailing the mechanism by which carotenoids can inhibit Aβ aggregation and disaggregation. 

## 7. Aβ Aggregation and DNA Damage

While we focused on the neuroprotective activity of carotenoids especially for their role of inhibition of Aβ aggregation, this property is also important due to its potential effect on Aβ mediated genome instability. Previous studies have demonstrated a relationship between Aβ aggregation and DNA damage [11]. This raised the question whether the carotenoid compounds that inhibit Aβ aggregation also inhibit or prevent the DNA damage [11]. Indeed, studies have found that carotenoids can prevent the DNA damage [115,116,117]. This activity can be related to the type of carotenoid: the pro-vitamin A and non-vitamin A groups of carotenoids. The pro-vitamin A can be found in β-carotene and β-cryptoxanthin and lycopene, lutein, and astaxanthin possess the non-vitamin A property. Lutein can prevent the DNA damage when they exposed lutein to ultraviolet radiation using human neuroblastoma and rat trachea epithelial cells [118,119]. Future studies should investigate the protective effect of carotenoids on genome integrity, which could be important for both neurodegeneration and cancer prevention. 

## 8. Conclusions and New Directions in Research

Carotenoids are natural products with a high therapeutic potential against neurodegenerative diseases, especially AD. For this reason, it is important to further explore comprehensively, mechanistic basis of carotenoids as a new source of therapeutic compounds in the fight against this devastating neurodegenerative disease. In this paper, we discussed the diversity of carotenoids found in nature and their high potential as therapeutic molecules involved in preventing multi-faceted toxic pathways of Aβ. Carotenoids are widely distributed in nature and exist in a variety of different types. However, only a few carotenoids have been characterized for their neuroprotective activity. We consider carotenoids as powerful natural compounds with nutraceutical and antioxidant properties that are critical in the fight against AD. Furthermore, most carotenoids can cross the blood-brain barrier. Future studies are in progress to evaluate the neuroprotective activity of carotenoids and their derivatives. 

## Figures and Tables

**Figure 1 ijms-20-05553-f001:**
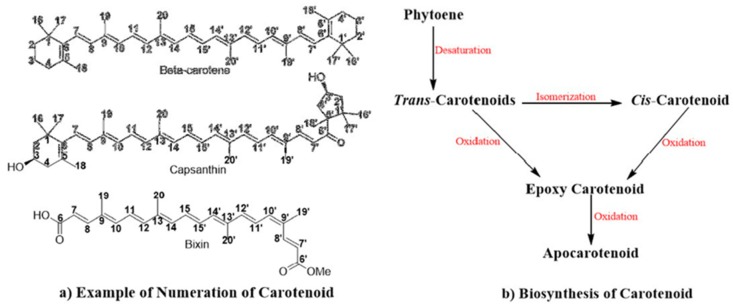
Numeration and biosynthesis of carotenoids: (**a**) Beta-carotene, capsanthin, and bixin are examples of how carotenoids and apocarotenoid can be numerated; (**b**) the biosynthesis of carotenoids starts with phytoenes that produce trans- and *cis*-carotenoids that can produce an epoxy carotenoid and finally an apocarotenoid.

**Figure 2 ijms-20-05553-f002:**
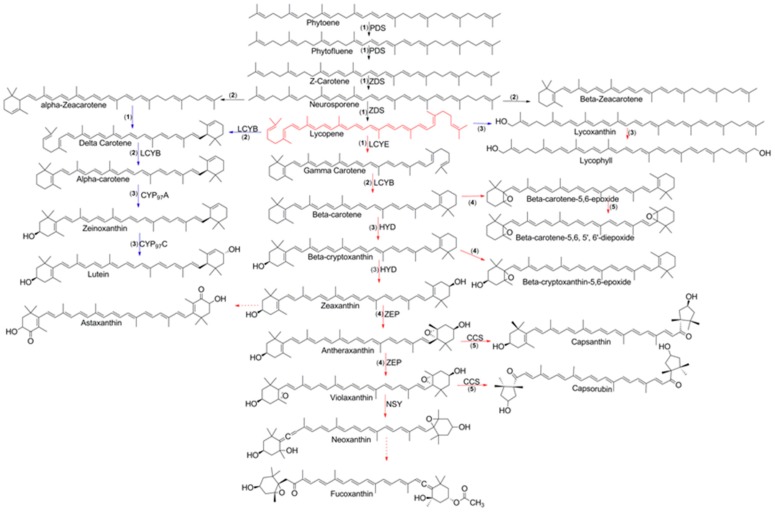
Biosynthesis of carotenoids and their five crucial reactions: desaturation (**1**), cyclization (**2**), hydroxylation (**3**), epoxidation (**4**), and epoxide-furanoxide rearrangement (**5**).

**Figure 3 ijms-20-05553-f003:**
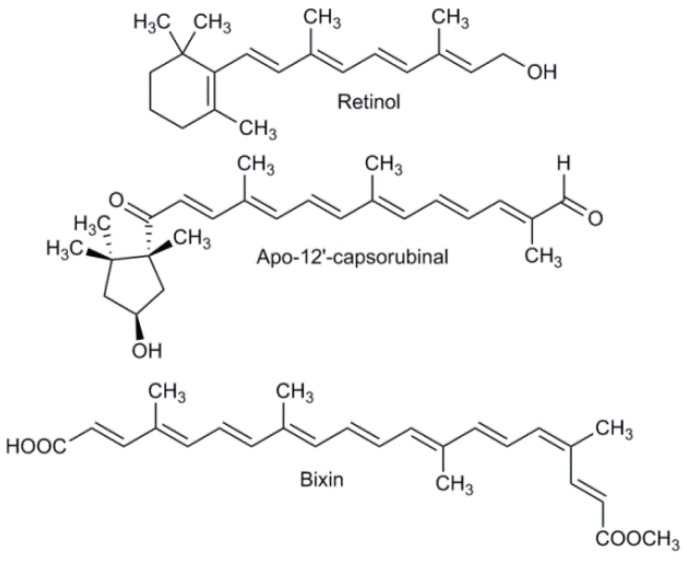
Examples of structures of some of the apocarotenoids.

**Figure 4 ijms-20-05553-f004:**
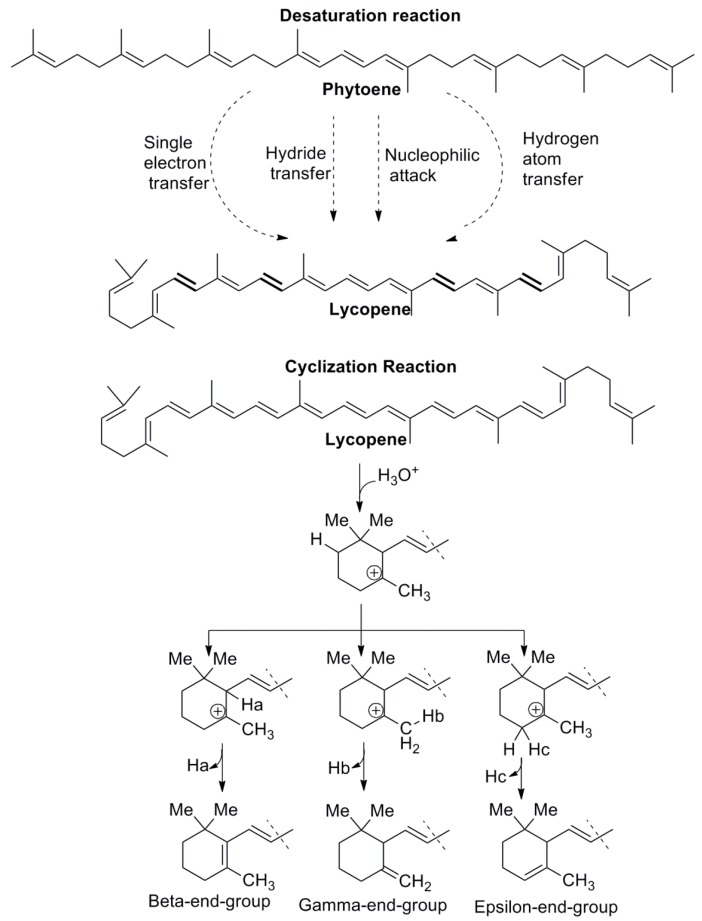
Desaturation and cyclization reaction.

**Figure 5 ijms-20-05553-f005:**
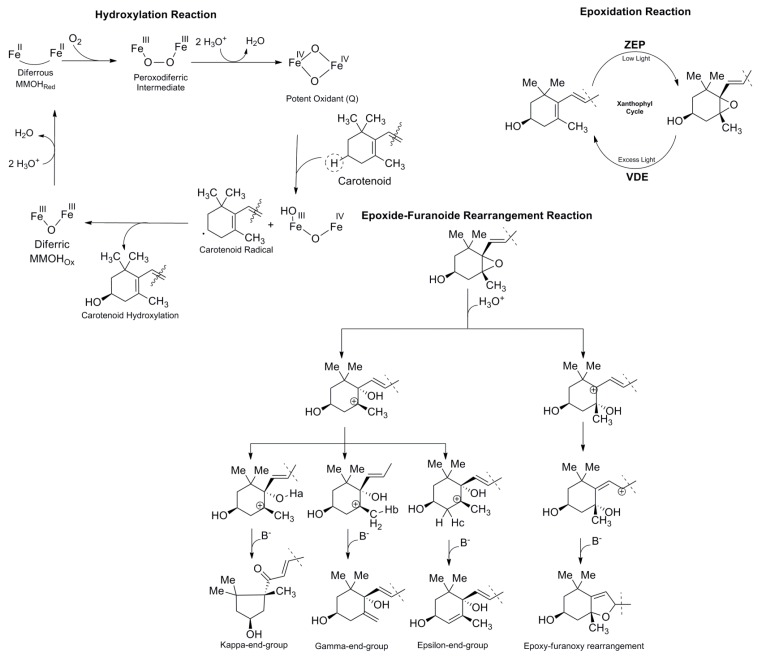
Hydroxylation, epoxidation, and epoxide-furanoide rearrangement reactions.

**Figure 6 ijms-20-05553-f006:**
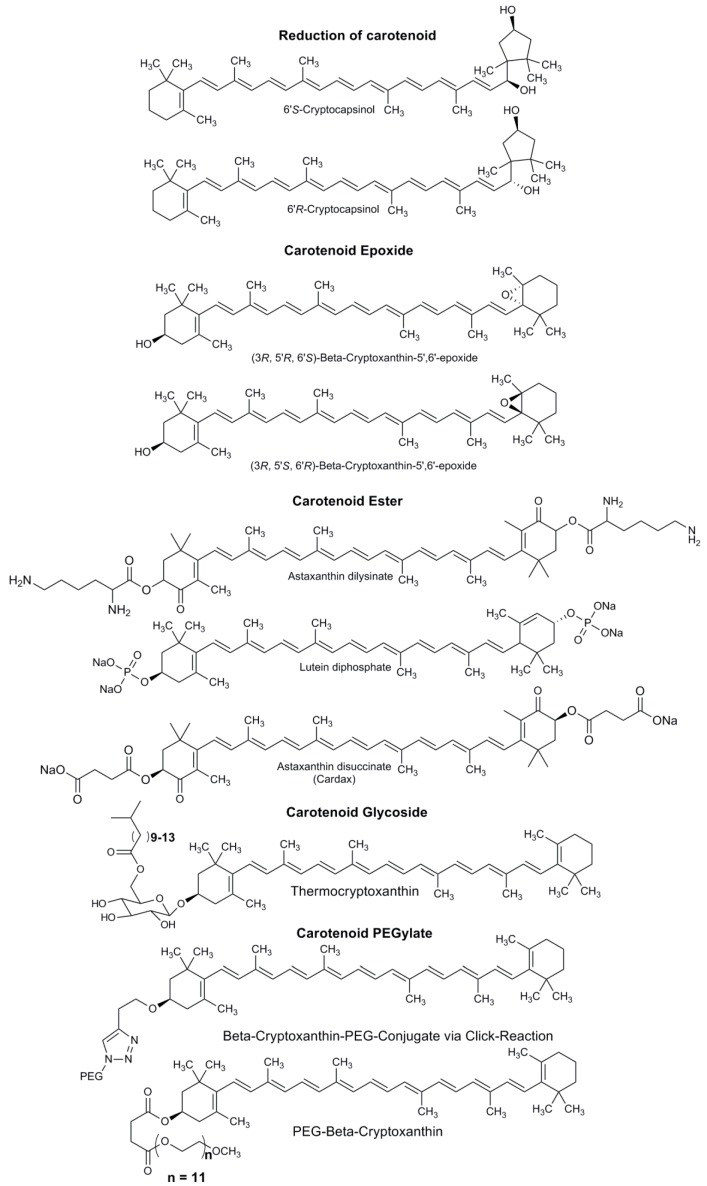
Types of carotenoid derivatives.

**Figure 7 ijms-20-05553-f007:**
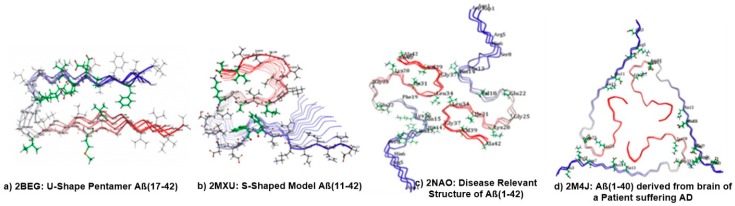
The Protein Data Bank (PDB) structures of four selected forms of Aβ fibrils are: (**a**) 2BEG including the Aβ_17–42_ region; (**b**) 2MXU is characterized by the Aβ_11–42_ region; (**c**) 2NAO is conformed by the Aβ_1–42_ region; (**d**) 2M4J contains the Aβ_1–40_ region.

**Figure 8 ijms-20-05553-f008:**
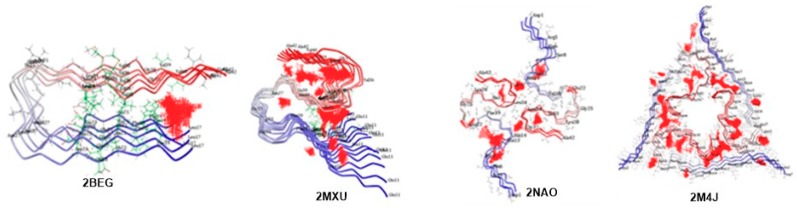
Probable receptor binding sites and alternative binding regions across different Aβ fibrils where ligand binding interactions can occur. Red dots correspond to the amino acid in the binding site.

**Figure 9 ijms-20-05553-f009:**
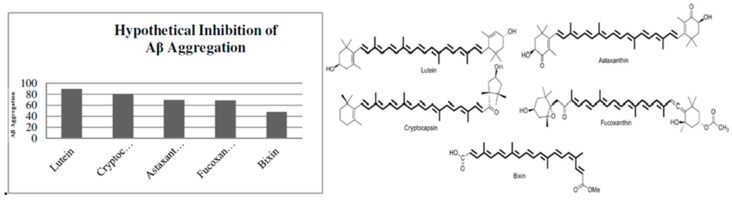
Hypothetical model of inhibition of Aβ aggregation by carotenoids and apocarotenoids. In this hypothetical inhibition of Aβ aggregation, lutein, a carotenoid with non-pro-vitamin A, showed the higher inhibition follow by cryptocapsin, a carotenoid with pro-vitamin A. After, astaxanthin and fucoxanthin, both of them are carotenoids with non-pro-vitamin A, and finally, Bixin, apocatenoid without pro-vitamin A.

**Figure 10 ijms-20-05553-f010:**
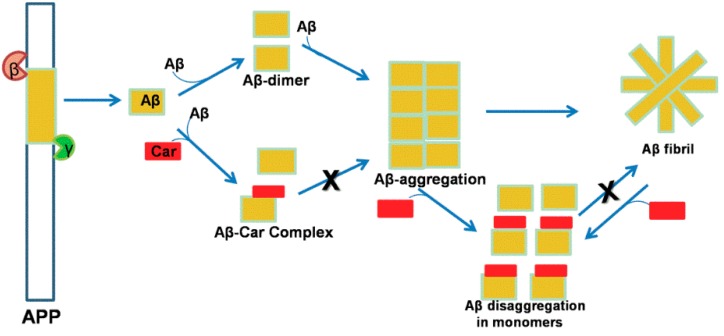
The hypothetical model to understand the role of carotenoids in the inhibition of Aβ_42_ formation. β- and γ-secretases are the two enzymes that cleavage the amyloid precursor protein (APP) to form Aβ peptide. The aggregation of this peptide pass to dimer follows to oligomer until Aβ fibril. Carotenoids may potentially inhibit the APP pathway where carotenoids have been demonstrated to inhibit Aβ dimer, also oligomer, and also the disaggregation of Aβ fibril. Color code: Aβ (yellow), carotenoid (red), β-secretase (brown) and γ-secretase (green).

**Table 1 ijms-20-05553-t001:** Interactions of ligands with the U-shaped pentamer Aβ_17–42_ using PDB ID: 2BEG.

Ligands	Highest -CDOCKER Energy (kcal/mol)	Highest -CDOCKER Interaction Energy (kcal/mol)	Amino Acid Residues Interacting with Aβ Chains	Common Residues Interacting Across all Ligands
**Astaxanthin**	−83.0351	−17.3682	Gly38, Val39	Ala21, Glu22 *, Val24 *, Gly25 *, Asp23 ^#^, Ser26 ^#^, Lys28 ^#^, Ile32, Gly33; Leu34 *, Gly38, Val39
**Bixin**	−85.044	−127.336	Glu22 *	
**Cryptocapsin**	−83.6518	−14.4395	Leu34 *, Ile32, Gly33, Gly33	
**Fucoxanthin**	−69.4873	−39.3977	Asp23 ^#^, Glu22, Ala21	
**Lutein**	−97.6659	−20.8785	Asp23 *, Val24 *^,#^, Gly25 *^,#^, Lys28 ^#^, Ser26 ^#^	

* Denotes main chain interactions and ^#^ specifies side chain interactions.

**Table 2 ijms-20-05553-t002:** Interactions of ligands with the S-shaped model structure of Aβ_11–42_ using PDB ID: 2MXU.

Ligands	Highest -CDOCKER Energy (kcal/mol)	Highest -CDOCKER Interaction Energy (kcal/mol)	Amino Acid Residues Interacting with Aβ Chains	Common Residues Interacting Across all Ligands
**Astaxanthin**	−36.1994	−60.0733	Val2, His14	Val2, His14 Ile32, Gly33, Leu34
**Bixin**	10.2499	−45.5219	Leu34, Gly33	
**Cryptocapsin**	−44.9219	−60.0403	Gly33, Ile32, Gly33	
**Fucoxanthin**	−40.2796	−60.6453	Leu34, His14	
**Lutein**	−60.6453	−50.7407	His14	

**Table 3 ijms-20-05553-t003:** Interactions of ligands with the structures of Aβ (1–42) using PDB ID: 2NAO-Disease-Relevant.

Ligands	Highest -CDOCKER Energy (kcal/mol)	Highest -CDOCKER Interaction Energy (kcal/mol)	Amino Acid Residues Interacting with Aβ Chains	Common Residues Interacting Across all Ligands
**Astaxanthin**	−76.0481	−29.5232	Ser8, Glu11, Gly9 Tyr10	Ser8, Glu11, Lys16
**Bixin**	−74.169	−104.431	Glu22, Glu11	
**Cryptocapsin**	−84.3430	−33.051	Lys16, Glu18	
**Fucoxanthin**	−75.0944	−37.7015	Lys16, His13	
**Lutein**	−88.1515	−30.5548	Ser8	

**Table 4 ijms-20-05553-t004:** Interactions of ligands with the structures of Aβ (1–40) derived from brain of a patient with AD using PDB ID: 2M4J.

Ligands	Highest -CDOCKER Energy (kcal/mol)	Highest -CDOCKER Interaction Energy (kcal/mol)	Amino Acid Residues Interacting with amyloid-β chains
**Astaxanthin**	−82.9702	−18.1365	-
**Bixin**	−22.6447	−16.4999	Lys28
**Cryptocapsin**	−101.782	−14.0123	Gly37
**Fucoxanthin**	−89.3804	−18.9031	Gly37
**Lutein**	−110.897	−16.1726	Gly37

**Table 5 ijms-20-05553-t005:** Summary of interactions of the five compounds with the Aβ (1–42) sequence. Color code: astaxanthin (red), bixin (blue), cryptocapsin (purple), fucoxanthin (green), and lutein (yellow).

Ligands	8	9	10	11	12	13	14	16	18	21	22	23	24	25	26	28	32	33	34	37	38	39
S	G	Y	E	V	H	H	K	V	A	E	D	V	G	S	K	I	G	L	G	G	V
**Astaxanthin**																						
**Bixin**																						
**Cryptocapsin**																						
**Fucoxanthin**																						
**Lutein**

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
