# Peer review of "Carotenoids as Novel Therapeutic Molecules Against Neurodegenerative Disorders: Chemistry and Molecular Docking Analysis"

_ijms, 2019, doi:10.3390/ijms20225553_

Round 1

Reviewer 1 Report

The manuscript added a new dimension of carotenoids used against neurodegenerative disease such as Alzheimer's disease. The molecular docking results showed interesting facts about anti-amyloidogenic potential and blood-brain barrier penetration potential. However, the abstract represent limited information about the study and required further addition of molecular docking results.

Author Response

Comment: The molecular docking results showed interesting facts about anti-amyloidogenic potential and blood-brain barrier penetration potential. However, the abstract represent limited information about the study and required further addition of molecular docking results.

Reply: The abstract is modified with more details on docking results

Abstract: Alzheimer’s disease (AD) is the most devastating neurodegenerative disorder that affects the aging population worldwide. Endogenous and exogenous factors are involved in triggering this complex and multifactorial disease, whose hallmark is amyloid-β (Aβ), formed by cleavage of amyloid precursor protein by β- and γ-secretase. While there is no definitive cure for AD to date, many neuroprotective natural products, such as polyphenol and carotenoid compounds, have shown promising preventive activity, as well as help in slowing down disease progression. In this article, we focus on the chemistry as well as structure of carotenoid compounds and their neuroprotective activity against Aβ aggregation using molecular docking analysis. In addition to examining the most prevalent anti-amyloidogenic carotenoid lutein, we studied cryptocapsin, astaxanthin, fucoxanthin, and the apocarotenoid bixin. Our computational structure-based drug design analysis and molecular docking simulation revealed important interactions between carotenoids and Aβ via hydrogen bonding and van der Waals interactions and shows that carotenoids are powerful anti-amyloidogenic molecules with a potential role in preventing AD, especially since most of them can cross the blood-brain barrier and are considered nutraceutical compounds. Our studies thus illuminate mechanistic insights on how carotenoids inhibit Aβ aggregation. The potential role of carotenoids as novel therapeutic molecules in treating AD and other neurodegenerative disorders are discussed.

Reviewer 2 Report

The manuscript by Johant Lakey-Beitia and collegues extensively discuss the diversity of carotenoids found in nature and their high potential as therapeutic molecules involved in preventing multi-faceted toxic pathways of Aß. In addition to inhibition of  Aß aggregation, various clinical studies have suggested  association of cartenoid consumption with lower risk of cardiovascular disease, cancer, and eye disease. The review is well written with discussion of data and results from different works in this field. However, the manuscript seems  bit long and might reduce the interest of readers. Having said that, I would not like to criticize the manuscript and would like to suggest accepting the manuscript in present form.

Author Response

Comment:  I would not like to criticize the manuscript and would like to suggest accepting the manuscript in the present form.

Reply: Minor modifications are done based on other reviewers. We made the executive manuscript to cover chemistry, docking models and neuroprotection as it serves for future researchers and students.